# Implicit Probabilistic Integrators for ODEs

**Onur Teymur**[*] **& Ben Calderhead**
Department of Mathematics
Imperial College London

**Han Cheng Lie & T.J. Sullivan**
Institute of Mathematics, Freie Universität Berlin;
& Zuse Institut Berlin

## Abstract

We introduce a family of implicit probabilistic integrators for initial value problems (IVPs), taking as a starting point the multistep Adams–Moulton method. The implicit construction allows for dynamic feedback from the forthcoming time-step, in contrast to previous probabilistic integrators, all of which are based on explicit methods. We begin with a concise survey of the rapidly-expanding field of probabilistic ODE solvers. We then introduce our method, which builds on and adapts the work of Conrad et al. (2016) and Teymur et al. (2016), and provide a rigorous proof of its well-definedness and convergence. We discuss the problem of the calibration of such integrators and suggest one approach. We give an illustrative example highlighting the effect of the use of probabilistic integrators—including our new method—in the setting of parameter inference within an inverse problem.

## 1  Set-up, motivation and context

We consider the common statistical problem of inferring model parameters $\theta$ from data $Y$. In a Bayesian setting, the parameter posterior is given by $p(\theta|Y) \propto p(Y|\theta)p(\theta)$. Suppose we have a regression model in which the likelihood term $p(Y|\theta)$ requires us to solve an ordinary differential equation (ODE). Specifically, for each datum, we have $Y_j = x(t_{Y_j}) + \varepsilon_j$ for some latent function $x(t)$ satisfying $\dot{x} = f(x, \theta)$ and vector of measurement errors $\varepsilon$ with spread parameter $\sigma$.

We can write the full model as $p(\theta, \sigma, x|Y) \propto p(Y|x, \sigma)p(x|\theta)p(\theta)p(\sigma)$. Since $x$ is latent, it is included as an integral part of the posterior model. This more general decomposition would not need to be considered explicitly in, say, a linear regression model for which $x = \theta_1 + \theta_2 t$; here we would simply have $p(x|\theta) = \delta_x(\theta_1 + \theta_2 t)$. In other words, given $\theta$ there is no uncertainty in $x$, and the model would reduce to simply $p(\theta, \sigma|Y) \propto p(Y|\theta, \sigma)p(\theta)p(\sigma)$.

In our case, however, $x$ is defined implicitly through the ODE $\dot{x} = f(x, \theta)$ and $p(x|\theta)$ is therefore no longer trivial. What we mean by this is that $x$ can only be calculated approximately and thus—following the central principle of *probabilistic numerical methods* (Hennig et al., 2015)—we assign to it a probability distribution representing our lack of knowledge about its true value. Our focus here is on initial value problems (IVPs) where we assume the initial value $X_0 \equiv x(0)$ is known (though an extension to unknown $X_0$ is straightforward). We thus have the setup

$$p(\theta, \sigma, x|Y, X_0) \propto p(Y|x, \sigma)p(x|\theta, X_0)p(\theta)p(\sigma). \tag{1}$$

For our purposes, the interesting term on the right-hand side is $p(x|\theta)$, where hereafter we omit $X_0$. In the broadest sense, our aim is to account as accurately as possible for the numerical error which is inevitable in the calculation of $x$, and to do this within a probabilistic framework by describing $p(x|\theta)$. We then wish to consider the effect of this uncertainty as it is propagated through (1), when performing inference on $\theta$ in an inverse problem setting. An experimental example in this context is considered in Section 3.

---

[*]Corresponding author: `o@teymur.uk`

## 1.1 Probabilistic numerical methods

Before we give a summary of the current state of probabilistic numerical methods (PN) for ODEs, we take a brief diversion. It is interesting to note that the concept of defining a distribution for $p(x|\theta)$ has appeared in the recent literature in different forms. For example, a series of papers (Calderhead et al., 2009; Dondelinger et al., 2013; Wang and Barber, 2014; Macdonald et al., 2015), which arose separately from PN in its modern form, seek to avoid solving the ODE entirely and instead replace it with a 'surrogate' statistical model, parameterised by $\phi$, with the primary motivation being to reduce overall computation. The central principle in these papers is to perform gradient matching (GM) between the full and surrogate models. A consequence of this framework is the introduction of a distribution $p(x|\theta, \phi)$ akin to the $p(x|\theta)$ appearing in (1). The aim is to approximate $x$ using statistical techniques, but there is no attempt to model the error itself—instead, simply an attempt to minimise the discrepancy between the true solution and its surrogate. Furthermore, the parameters $\phi$ of the surrogate models proposed in the GM framework are fitted by conditioning on data $Y$, meaning $p(x|\theta, \phi)$ needs to be viewed as a data-conditioned posterior $p(x|\theta, \phi, Y)$. In our view this is problematic, since where the uncertainty in a quantity of interest arises solely from the inexactness of the *numerical* methods used to calculate it, inference over that quantity should not be based on data that is the outcome of an experiment. The circularity induced in (1) by $Y$-conditioning is clear.

The fundamental shift in thinking in the papers by Hennig and Hauberg (2014) and Chkrebtii et al. (2016), building on Skilling (1991), and then followed up and extended by Schober et al. (2014), Conrad et al. (2016), Teymur et al. (2016), Kersting and Hennig (2016), Schober et al. (2018) and others is that of what constitutes 'data' in the algorithm used to determine $p(x|\theta)$. By contrast to the GM approach, the experimental data $Y$ is *not* used in constructing this distribution. Though the point has already been made tacitly in some of these works, we argue that this constitutes the key difference in philosophy. Instead, we should strive to quantify the numerical uncertainty in $x$ first, then propagate this uncertainty via the data likelihood to the Bayesian inversion employed for inferring $\theta$. This is effectively direct probabilistic modelling of the numerical error and is the approach taken in PN.

How then is $x$ inferred in PN? The common thread here is that a discrete path $Z \equiv Z_{1:N}$ is generated which numerically approximates $X \equiv X_{1:N}$—the discretised version of the true solution $x$—then 'model interrogations' (Chkrebtii et al., 2016) $F := f(Z, \theta)$ are thought of as a sort of numerical data and $x$ is inferred based on these. Done this way, an entirely model-based description of the uncertainty in $x$ results, with no recourse to experimental data $Y$.

## 1.2 Sequential inference

Another central feature of PN solvers from Chkrebtii et al. (2016) onward is that of treating the problem sequentially, in the manner of a classic IVP integrator. In all of the GM papers, and indeed in Hennig and Hauberg (2014), $X$ is treated as a block – inferred all at once, given data $Y$ (or, in Hennig and Hauberg, $F$). This necessarily limits the degree of feedback possible from the dynamics of the actual ODE, and in a general IVP this may be the source of significant inaccuracy, since errors in the inexact values $Z$ approximating $X$ are amplified by the ODE itself. In a sequential approach, the numerical data is not a static pre-existing object as the true data $Y$ is, but rather is generated as we go by repeatedly evaluating the ODE at a sequence of input ordinates. Thus it is clear that the numerical data generated at time $t$ is affected by the inferred solution at times before $t$. This iterative information feedback is qualitatively much more like a standard IVP solver than a block inference approach and is similar to the principle of statistical filtering (Särkkä, 2013).

We now examine the existing papers in this area more closely, in order to give context to our own contribution in the subsequent section. In Chkrebtii et al. (2016) a Gaussian process (GP) prior is jointly placed over $x$ and its derivative $\dot{x}$, then at step $i$ the current GP parameters are used to predict a value for the state at the next step, $Z_{i+1}$. This is then transformed to give $F_{i+1} \equiv f(Z_{i+1}, \theta)$. The modelling assumption now made is that this value is distributed around the true value of the derivative $\dot{X}_{i+1}$ with Gaussian error. On this basis the new datum is assimilated into the model, giving a posterior for $(x, \dot{x})$ which can be used as the starting prior in the next step. This approach does not make direct use of the sequence $Z$; rather it is merely generated in order to produce the numerical data $F$ which is then compared to the prior model in derivative space. The result is a distributional Gaussian posterior over $x$ consistent with the sequence $F$.

Conrad et al. (2016) take a different approach. Treating the problem in a discrete setting, they produce a sequence $Z$ of values approximating $X$, with $F_{i+1} \equiv f(Z_{i+1}, \theta)$ constituting the data and $Z_{i+1}$ calculated from the previous values $Z_i$ and $F_i$ by employing some iterative relation akin to a randomised version of a standard IVP solver. Note that there is no attempt to continuously assimilate the generated values into the model for the unknown $X$ or $\dot{X}$ during the run of the algorithm. Instead, the justification for the method comes *post hoc* in the form of a convergence theorem bounding the maximum expected squared-error $\max_i \mathbb{E}||Z_i - X_i||^2$. An extension to multistep methods—in which $Z_{i+1}$ is allowed to depend on multiple past values $F_{\leq i}$—is introduced in Teymur et al. (2016) and utilises the same basic approach. Various extensions and generalisations of the theoretical results in these papers are given in Lie et al. (2017), and a related idea in which the step-size is randomised is proposed by Abdulle and Garegnani (2018).

This approach is intuitive, allowing for modified versions of standard algorithms which inherit known useful properties, and giving provable expected error bounds. It is also more general since it allows for non-parametric posterior distributions for $x$, though it relies on Monte Carlo sampling to give empirical approximations to it. Mathematically, we write

$$p(Z|\theta) = \int p(Z, F|\theta)\, \mathrm{d}F = \int \left[ \prod_{i=0}^{N-1} p(F_i|Z_i, \theta)p(Z_{i+1}|Z_i, F_{\leq i}) \right] \mathrm{d}F. \qquad (2)$$

Here, $Z \equiv Z_{1:N}$ is the approximation to the unknown discretised solution function $X \equiv X_{1:N}$, and each $F_i \equiv f(Z_i, \theta)$ is a piece of numerical data. We use $F_{\leq i}$ to mean $(F_i, F_{i-1}, F_{i-2}, \dots)$. Using the terminology of Hennig et al. (2015), the two constituent components of the telescopic decomposition in the right-hand side of (2) correspond to the 'decision rule' (how the algorithm generates a new data-point $F_i$) and the 'generative model' (which encodes the likelihood model for $Z$) respectively. Note that, from a statistical viewpoint, the method explicitly defines a distribution over numerical solutions $Z$ rather than an uncertainty centred around the true solution $x$ (or $X$). The relationship of the measure over $Z$ to that over $X$ is then guaranteed by the convergence analysis.

The term $p(F_i|Z_i, \theta)$ is taken in both Conrad et al. (2016) and Teymur et al. (2016) to be simply a deterministic transformation; this could be written in distributional form as $\delta_{F_i}(f(Z_i, \theta))$. The term $p(Z_{i+1}|Z_i, F_{\leq i})$ is given by Conrad et al. as a Gaussian centred around the output $Z_{i+1}^{\mathrm{det}}$ of any deterministic single step IVP solver, with variance scaled in accordance with the constraints of their theorem. Teymur et al. introduce a construction for this term which permits conditioning on multiple previous $F_i$'s and has mean equivalent to the multistep Adams–Bashforth method. They give the corresponding generalised convergence result. Their proof is also easily verified to be valid for implicit multistep methods—a result we appeal to later—though the specific implicit integrator model they suggest is methodologically inconsistent, for reasons we will explain in Section 2.

In all of the approaches described so far, Monte Carlo sampling is required to marginalise $F$ and thereby calculate $p(Z|\theta)$. This constitutes an appreciable computational overhead. A third approach, related to stochastic filtering, is presented in Schober et al. (2014), Kersting and Hennig (2016) and Schober et al. (2018). These papers develop a framework which does not rely on sampling, but instead makes the simplifying assumption that all distributions are Gaussian, and propagates the uncertainty from step to step using the theory of Kalman filtering (Särkkä, 2013). This is an alternative settlement to the accuracy/computational cost trade-off, a point which is acknowledged in those papers.

For the sake of comparison, we can loosely rewrite their general approach in our notation as follows:

$$p(x|\theta) = \int \left[ \prod_i p(\tilde{Z}_{i+1}|x^{[i]}, F_{0:i})p(F_{i+1}|\tilde{Z}_{i+1}, \theta)p(x^{[i+1]}|\tilde{Z}_{i+1}, F_{i+1}) \right] \mathrm{d}F\, \mathrm{d}\tilde{Z}, \qquad (3)$$

where we write $x^{[i]}$ instead of $Z_i$ to emphasise that this represents an $i$-times updated model for the continuous solution $x$, rather than the $i$'th iteration of an algorithm which generates an approximation to the discrete $X_i$. This algorithm predicts a value for the new state $\tilde{Z}_{i+1}$ from the current model and all previous data, then generates a data point based on that prediction, and then updates the model based on this new datum. Note that all distributions in this framework are Gaussian, to permit fast filtering, and as a result the non-linearities in $f$ are effectively linearised, and any numerical method which produces non-Gaussian errors has Gaussians fitted to them anyway.

This filtering approach is interesting because of the earlier-stated desideratum of maximising the degree of feedback from the ODE dynamics to the solver. The predict-evaluate-update approach

suggested by (3) means that information from the ODE function at the *next* time step $t_{i+1}$ is fed back into the procedure at each step, unlike in other methods which only predict forwards. In numerical analysis this is typically a much-desired feature, leading to methods with improved stability and accuracy. However, it is still a three-part procedure, analogous for example to paired Adams–Bashforth and Adams–Moulton integrators used in PEC mode (Butcher, 2008). This connection is referred to in Schober et al. (2018).

## 2 Our proposed method

We now propose a different, novel sequential procedure which also incorporates information from the ODE at time step $t_{i+1}$ but does so directly. This produces a true implicit probabilistic integrator, without a subtle inconsistency present in the method suggested by Teymur et al. (2016). There, the analogue of (2) defines a joint Gaussian distribution over $Z_{i+1}$ and $F_{i+1}$ (the right-hand component, with $F_{\leq i}$ replaced by $F_{\leq (i+1)}$) but then generates $F_{i+1}$ by passing $Z_{i+1}$ through the function $f$ (the left hand component). This gives two mutually-incompatible meanings to $F_{i+1}$, one linearly and one non-linearly related to $Z_{i+1}$. Our proposed method fixes this problem. Indeed, we specifically exploit the difference in these two quantities by separating them out and directly penalising the discrepancy between them.

To introduce the idea we consider the one-dimensional case first, then later we generalise to a multi-dimensional context. We first note that unlike in the explicit randomised integrators of Conrad et al. (2016) and Teymur et al. (2016), we do not have access to the exact deterministic Adams–Moulton predictor, to which we could then add a zero-mean perturbation. An alternative approach is therefore required. Consider instead the following distribution which directly advances the integrator one step and depends only the current point:

$$p(Z_{i+1} = z | Z_i, \theta, \eta) \propto g(r(z), \eta). \tag{4}$$

Here, $r(z)$ is a positive discrepancy measure *in derivative space* defined in the next paragraph, and $g$ is an $\eta$-scaled functional transformation which ensures that the expression is a valid probability distribution in the variable $z$.

A concrete example will illuminate the definition. Consider the simplest implicit method, backward Euler. This is defined by the relation $Z_{i+1} = Z_i + hF_{i+1}$ and typically can only be solved by an iterative calculation, since $F_{i+1} \equiv f(Z_{i+1}, \theta)$ is of course unknown. If the random variable $Z_{i+1}$ has value $z$, then we may express $F_{i+1}$ as a function of $z$. Specifically, we have $F_{i+1}(z) = h^{-1}(z - Z_i)$. The discrepancy $r(z)$ between the value of $F_{i+1}(z)$ and the value of $f(z, \theta)$ can then be used as a measure of the error in the linear method, and penalised. This is equivalent to penalising the difference between the two different expressions for $F_{i+1}$ arising from the previously-described naive extension of (2) to the implicit case. We write

$$p(Z_{i+1} = z | Z_i, \theta, \eta) = K^{-1} \exp\left(-\tfrac{1}{2}\eta^{-2}\left(h^{-1}(z - Z_i) - f(z, \theta)\right)^2\right). \tag{5}$$

Comparing (4) and (5), $r(z)$ is the expression $h^{-1}(z - Z_i) - f(z, \theta)$, and $g$ is in this case the transformation $u \mapsto \exp(-u^2/2\eta^2)$. This approach directly advances the solver in a single leap, without collecting explicit numerical data as in previous approaches. It is in general non-parametric and requires either sampling or approximation to be useful (more on which in the next section). Since $f$ is in general non-linear, it follows that $r$ is non-linear too. It then follows that the density in equation (5) does *not* result in a Gaussian measure, despite $g$ being a squared-exponential transformation. The generalisation to higher order implicit linear multistep methods of Adams–Moulton (AM) type, having the form $Z_{i+1} = Z_i + h\sum_{j=-1}^{s-1}\beta_j f(Z_{i-j}, \theta)$, for AM coefficients $\beta_j$, follows as

$$p(Z_{i+1} = z | Z_{\leq i}, \theta, \eta) = \frac{1}{K} \exp\left(-\frac{1}{2\eta^2}\left(\frac{h^{-1}(z - Z_i) - \sum_{j=0}^{s-1}\beta_j F_{i-j}}{\beta_{-1}} - f(z, \theta)\right)^2\right). \tag{6}$$

### 2.1 Mathematical properties of the proposed method

The following analysis proves the well-definedness and convergence properties of the construction proposed in Section 2. First we show that the distribution (6) is well-defined and proper, by proving

the finiteness and strict positivity of the normalising constant $K$. We then describe conditions on the $h$-dependence of the scaling parameter $\eta$, such that the random variables $\xi_i$ in (8) satisfy the hypotheses of Theorem 3 in Teymur et al. (2016). In particular, the convergence of our method follows from the Adams–Moulton analogue of that result.

Denote by $\Psi^h_{\theta,s} : \mathbb{R}^{d \times s} \to \mathbb{R}^d$ the deterministic map defined by the $s$-step Adams–Moulton method. For example, the implicit map associated with the backward Euler method—the 'zero step' AM method—for a fixed parameter $\theta$ is $\Psi^h_{\theta,0}(Z_i) = Z_i + hf(\Psi^h_{\theta,0}(Z_i), \theta)$. More generally, the map associated with the $s$-step AM method is

$$\Psi^h_{\theta,s}(Z_{i-s+1:i}) = Z_i + h\Big[\beta_{-1}f\big(\Psi^h_{\theta,s}(Z_{i-s+1:i}), \theta\big) + \sum_{j=0}^{s-1}\beta_j f(Z_{i-j}, \theta)\Big], \qquad (7)$$

where $Z_{i-s+1:i} \equiv (Z_i, Z_{i-1}, \ldots, Z_{i-s+1})$, and the $\beta_j \in \mathbb{R}_+$ are the Adams–Moulton coefficients. Note that $\Psi^h_{\theta,s}(Z_{i-s+1:i})$ represents the deterministic Adams–Moulton estimate for $Z_{i+1}$. Given a probability space $(\Omega, \mathcal{F}, \mathbb{P})$, define for every $i \in \mathbb{N}$ the random variable $\xi^h_i : \Omega \to \mathbb{R}^d$ according to

$$Z_{i+1} = \Psi^h_{\theta,s}(Z_{i-s+1:i}) + \xi^h_i. \qquad (8)$$

The relationship between the expressions (6) and (8) is addressed in part *(i)* of the following Theorem, the proof of which is given in the supplementary material accompanying this paper.

**Theorem.** *Assume that the vector field $f(\cdot, \theta)$ is globally Lipschitz with Lipschitz constant $L_{f,\theta} > 0$. Fix $s \in \mathbb{N} \cup \{0\}$, $Z_{i-s+1:i} \in \mathbb{R}^{d \times s}$, $\theta \in \mathbb{R}^q$, and $0 < h < (L_{f,\theta}\beta_{-1})^{-1}$. If $\eta = kh^\rho$ for some $k > 0$ independent of $h$ and $\rho \geq -1$, then the following statements hold:*

*(i)* *The function defined in (6) is a well-defined probability density.*

*(ii)* *For every $r \geq 1$, there exists a constant $0 < C_r < \infty$ that does not depend on $h$, such that for all $i \in \mathbb{N}$, $\mathbb{E}[\|\xi_i\|^r] \leq C_r h^{(\rho+1)r}$.*

*(iii)* *If $\rho \geq s + \frac{1}{2}$, the probabilistic integrator defined by (6) converges in mean-square as $h \to 0$, at the same rate as the deterministic $s$-step Adams–Moulton method.*

## 2.2 Multi-dimensional extension

The extrapolation part of any linear method operates on each component of a multi-dimensional problem separately. Thus if $Z = (Z^{(1)}, \ldots, Z^{(d)})^T$, we have $Z^{(k)}_{i+1} = Z^{(k)}_i + h\sum_j \beta_j F^{(k)}_{i-j}$ for each component $k$ in turn. Of course, this is not true of the transformation $Z_{i+1} \mapsto F_{i+1} \equiv f(Z_{i+1})$, except in the trivial case where $f$ is linear in $z$; thus in (2), the right-hand distribution is componentwise-independent while the left-hand one is not. All previous sequential PN integrators have treated the multi-dimensional problem in this way, as a product of one-dimensional relations.

In our proposal it does not make sense to consider the system of equations component by component, due to the presence of the non-linear $f(z, \theta)$ term, which appears as an intrinsic part of the step-forward distribution $p(Z_{i+1}|Z_{\leq i}, \theta, \eta)$. The multi-dimensional analogue of (6) should take account of this and be defined over all $d$ dimensions together. For vector-valued $z, Z_k, F_k$, we therefore define

$$p(Z_{i+1}|Z_{\leq i}, \theta, H) \propto \exp\left\{-\tfrac{1}{2}r(z)^T H^{-1} r(z)\right\}. \qquad (9)$$

where $r(z) = \beta^{-1}_{-1}(h^{-1}(z - Z_i) - \sum_{j=0}^{s-1}\beta_j F_{i-j}) - f(z, \theta)$ is now a $d \times 1$ vector of discrepancies in derivative space, and $H$ is a $d \times d$ matrix encoding the solver scale, generalising $\eta$. Straightforward modifications to the proof give multi-dimensional analogues to the statements in the Theorem.

## 2.3 Calibration and setting $H$

The issue of calibration of ODE solvers is addressed without consensus in every treatment of this topic referenced in Section 1. The approaches can broadly be split into those of 'forward' type, in which there is an attempt to directly model what the theoretical uncertainty in a solver step should be and propagate that through the calculation; and those of 'backward' type, where the uncertainty scale is somehow matched *after* the computation to that suggested by some other indicator. Both of these have shortcomings, the former due to the inherent difficulty of explicitly describing the error, and the latter because it is by definition less precise. One major stumbling block is that it is in general a challenging problem to even *define* what it means for an uncertainty estimate to be well-calibrated.

In the present paper, we require a way of setting $H$. We proceed by modifying and generalising an idea from Conrad et al. (2016) which falls into the 'backward' category. There, the variance of the step-forward distribution $\mathrm{Var}(Z_{i+1}|\cdots)$ is taken to be a matrix $\Sigma_Z = \alpha h^\rho \mathbb{I}_d$, with $\alpha$ determined by a scale-matching procedure that ensures the integrator outputs a global error scale in line with expectations. We refer the reader to the detailed exposition of this procedure in Section 3.1 of that paper. Furthermore, the convergence result from Teymur et al. (2016) implies that, for the probabilistic $s$-step Adams–Bashforth integrator, the exponent $\rho$ should be taken to be $2s + 1$.

In our method, we are not able to relate such a matrix $\Sigma_Z$ directly to $H$ because from the definition (9) it is clear that $H$ is a scaling matrix for the spread of the *derivative* $F_{i+1}$, whereas $\Sigma_Z$ measures the spread of the *state* $Z_{i+1}$. In order to transform to the correct space without linearising the ODE, we apply the multivariate delta method (Oehlert, 1992) to give an approximation for the variance of the transformed random variable, and set $H$ to be equal to the result. Thus

$$\begin{aligned} H = \mathrm{Var}(f(Z_{i+1})) &\approx J_f(\mathbb{E}(Z_{i+1}))\Sigma_Z J_f(\mathbb{E}(Z_{i+1}))^T \\ &= \alpha h^\rho J_f(\mathbb{E}(Z_{i+1})) J_f(\mathbb{E}(Z_{i+1}))^T, \end{aligned} \tag{10}$$

where $J_f$ is the Jacobian of $f$. The mean value $\mathbb{E}(Z_{i+1})$ is unknown, but we can use an explicit method of equal or higher order to compute an estimate $Z_{i+1}^{\mathrm{AB}}$ at negligible cost, and use $J_f(Z_{i+1}^{\mathrm{AB}})$ instead, under the assumption that these are reasonably close. Remember that we are roughly calibrating the method so some level of approximation is unavoidable. This comment applies equally to the case where the Jacobian is not analytically available and is estimated numerically. Such approximations do not affect the fundamental convergence properties of the algorithm, since they do not affect the $h$-scaling of the stepping distribution. We also note that we are merely matching variances/spread parameters and nowhere assuming that the distribution (9) is Gaussian. This idea bears some similarity to the original concept in Skilling (1993), where a scalar 'stiffness constant' is used in a similar way to transform the uncertainty scale from solution space to derivative space.

We now ascertain the appropriate $h$-scaling for $H$ by setting the exponent $\rho$. The condition required by the univariate analysis in this paper is that $\eta = kh^\rho$; part *(iii)* of the Theorem shows that we require $\rho \geq s + \frac{1}{2}$, where $s$ is the number of steps in the corresponding AM method.[2] Choosing $\rho = s + \frac{1}{2}$ —an approach supported by the numerical experiments in Section 3—the backwards Euler method ($s = 0$) requires $\rho = \frac{1}{2}$. The multidimensional analogue of the above condition is $H = Qh^{2\rho}$ for an $h$-independent positive-definite matrix $Q$. Since $J_f$ is independent of $h$, this means we must set $\Sigma_Z$ to be proportional to $h^{2(s+\frac{1}{2})}$, and thus we have $H = \alpha h^{2s+1} J_f(\mathbb{E}(Z_{i+1})) J_f(\mathbb{E}(Z_{i+1}))^T$.

Our construction has the beneficial consequence of giving a non-trivial cross-correlation structure to the error calibration matrix $H$, allowing a richer description of the error in multi-dimensional problems, something absent from previous approaches. Furthermore, it derives this additional information via direct feedback from the ODE, which we have shown is a desirable attribute.

## 2.4 Reducing computational expenditure

In the form described in the previous section, our algorithm results in a non-parametric distribution for $Z_{i+1}$ at each step. With this approach, a description of the uncertainty in the numerical method can only be evaluated by a Monte Carlo sampling procedure at every iteration. Even if this sampling is performed using a method well-suited to targeting distributions close to Gaussian—we use a modified version of the pre-conditioned Crank–Nicolson algorithm proposed by Cotter et al. (2013)—there is clearly a significant computational penalty associated with this.

The only way to avoid this penalty is by reverting to distributions of standard form, which are easy to sample from. One possibility is to approximate (6) by a Gaussian distribution—depending on how this approximation is performed the desideratum of maintaining information feedback from the future dynamics of the target function can be maintained. For example, a first order Taylor expansion of $f(z) \approx f(Z_i) + J_f(Z_i)(z - Z_i)$, when substituted into $r(z)$ as defined in (9), gives an approximation $\tilde{r}(z)$ which is linear in $z$. This yields a non-centred Gaussian when transformed into a probability measure as in (9). Defining $\Gamma \equiv (h\beta_{-1}\mathbb{I}_d)^{-1} - J_f(Z_i)$ and $w \equiv f(Z_i, \theta) + \beta_{-1}^{-1}(\sum_{j=0}^{s-1} \beta_j F_{i-j})$,

some straightforward algebra gives the moments of the approximating Gaussian measure for the next step as $\mu = Z_i + \Gamma^{-1}w$ and $\mathrm{Var} = \alpha h^{2s+1}\Gamma^{-1}J_f J_f^T \Gamma^{-T}$. We note that this procedure is merely to facilitate straightforward sampling—though $\tilde{r}(z)$ is linear in $z$, the inclusion of the first additional term from the Taylor expansion means that information about the non-linearity (in $z$) of $f$ are still incorporated to second order, and the generated solution $Z$ is not jointly Gaussian across time steps $i$. Furthermore, since $\Gamma^{-1}$ is order 1 in $h$, this approximation does not impact the global convergence of the integrator, as long as $H$ is set in accordance with the principles described in Section 2.3. This method of solving implicit integrators by linearising them in $f$ is well-known in classical numerical analysis, and the resulting methods are sometimes called *semi-implicit* methods (Press et al., 2007).

## 3 Experimental results

We illustrate our new algorithm by considering the case of a simple inverse problem, the FitzHugh–Nagumo model discussed in Ramsay et al. (2007) and subsequently considered in a several papers on this topic. This is a two-dimensional non-linear dynamical system with three parameters $\theta = (\theta_1, \theta_2, \theta_3)$, the values of which ($\theta_1 = 0.2, \theta_2 = 0.2, \theta_3 = 3.0$) are chosen to produce periodic motion. With the problem having a Bayesian structure, we write down the posterior as

$$p(\theta, Z|Y) \propto p(Y|Z, \sigma)p(Z|\theta, \xi)p(\theta)p(\xi). \qquad (11)$$

This expression recalls (1), but with $Z$ substituting for $x$ as described in Section 1.2. We write $p(Z|\theta, \xi)$ to emphasise that the trajectory $Z$ depends on the sequence of random perturbations $\xi_{0:N}$. For simplicity we use the known value of $\sigma$ throughout, so do not include it in the posterior model.

Conrad et al. (2016) remark on the biasing effect on the posterior distribution for $\theta$ of naively evaluating the forward model using a standard numerical method. They showed that their probabilistic integrator returns wider posteriors, preventing misplaced overconfidence in an erroneous estimate. We now extend these experiments to our new method. In passing, we note interesting recent theoretical developments discussing the quantitative effect on posterior inference of randomised forward models, presented in Lie et al. (2018).

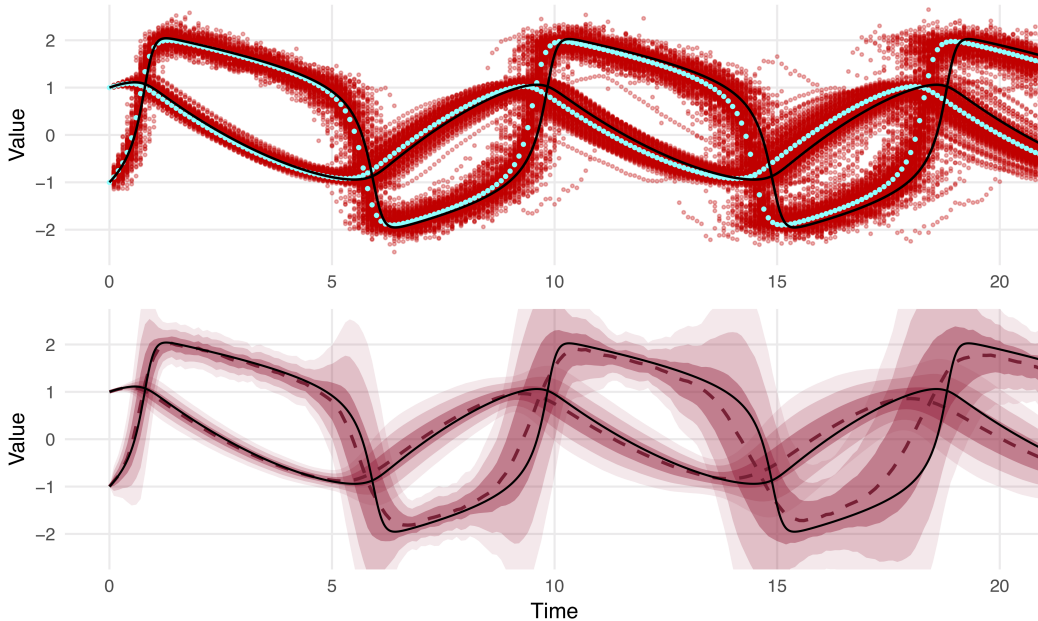

Figure 1: 500 Monte Carlo repetitions of the probabilistic backward Euler (AM0) method applied to the FitzHugh–Nagumo model with $h = 0.1$ and $0 \leq t \leq 20$. The approximation from Section 2.4 is used and $\alpha^*_{\mathrm{AM0}} = 0.2$. The upper pane plots the ensemble of discrete trajectories, with the path of the deterministic backward Euler method in light blue. The lower pane is based on the same data, this time summarised. The ensemble mean is shown dashed, and $1\sigma$, $2\sigma$ and $3\sigma$ intervals are shown shaded, with reference solution in solid black.

## 3.1 Calibration

The first task is to infer the appropriate value for the overall scaling constant $\alpha$ for each method, to be used in setting the matrix $H$ in (10). As in Conrad et al. (2016), we calculate a value $\alpha^*$ that maximises the agreement between the output of the probabilistic integrator and a measure of global error from the deterministic method, and then fix and proceed with this value.

For each of several methods $M$, $\alpha_M^*$ was calculated for a range of values of $h$ and was close to constant throughout, suggesting that the $h$-scaling advocated in Section 2.3 (ie. taking the equality in the bound in part *(iii)* of the Theorem) is the correct one. This point has not been specifically addressed in previous works on this subject. The actual maxima $\alpha_M^*$ for each method are different and further research is required to examine whether a relationship can be deduced between these values and some known characteristic of each method, such as number of steps $s$ or local error constant of the underlying method. Furthermore, we expect these values to be problem-dependent. In this case, we found $\alpha_{AB1}^* \approx 0.2$, $\alpha_{AB2}^* \approx 0.1$, $\alpha_{AB3}^* \approx 0.2$, $\alpha_{AM0}^* \approx 0.2$, $\alpha_{AM1}^* \approx 0.05$, $\alpha_{AM2}^* \approx 0.05$.

Having calibrated the probabilistic integrator, we illustrate its typical output in Figure 1: the top pane plots the path of 500 iterations of the probabilistic backward Euler method run at $\alpha = \alpha_{AM0}^* = 0.2$. We plot the discrete values $Z_{1:N}$ for each repetition, without attempting to distinguish the trajectories from individual runs. This is to stress that each randomised run (resulting from a different instantiation of $\xi$) is not intended to be viewed as a 'typical sample' from some underlying continuous probability measure, as in some other probabilistic ODE methods, but rather that collectively they form an ensemble from which an empirical distribution characterising discrete-time solution uncertainty can be calculated. The bottom pane plots the same data but with shaded bands representing the $1\sigma$, $2\sigma$ and $3\sigma$ intervals, and a dotted line representing the empirical mean.

## 3.2 Parameter inference

We now consider the inverse problem of inferring the parameters of the FitzHugh–Nagumo model in the range $t \in [0, 20]$. We first generate synthetic data $Y$; 20 two-dimensional data-points collected at times $t_Y = 1, 2, \dots, 20$ corrupted by centred Gaussian noise with variance $\sigma = (0.01) \cdot \mathbb{I}_2$. We then treat the parameters $\theta$ as unknown and run an MCMC algorithm—Adaptive Metropolis Hastings (Haario et al., 2001)—to infer their posterior distribution.

In Conrad et al. (2016), the equivalent algorithm performs multiple repetitions of the forward solve at each step of the outer MCMC (each with a different instantiation of $\xi$) then marginalises $\xi$ out to form an expected likelihood. This is computationally very expensive; in our experiments we find that for the MCMC to mix well, many tens of repetitions of the forward solve are required *at each step*.

Instead we use a Metropolis-within-Gibbs scheme where at MCMC iteration $k$, a candidate parameter $\theta^*$ is proposed and accepted or rejected having had its likelihood calculated using the same sample $\xi_{0:N}^{[k]}$ as used in the current iteration $k$. If accepted as $\theta^{[k+1]}$, a new $\xi_{0:N}^{[k+1]}$ can then be sampled and the likelihood value recalculated ready for the next proposal. The proposal at step $k + 1$ is then compared to this new value. Pseudo-code for this algorithm is given in the supplementary material.

Our approach requires that $p(Z|\theta, \xi)$ be recalculated exactly once for each time a new parameter value $\theta^*$ is accepted. The cost of this strategy is therefore bounded by twice the cost of an MCMC operating with a deterministic integrator—the bound being achieved only in the scenario that all proposed moves $\theta^*$ are accepted. Thus the algorithm, in contrast to the calibration procedure (which is relatively costly but need only be performed once), has limited additional computational overhead compared to the naive approach using a classical method.

Figure 2 shows kernel density estimates approximating the posterior distribution of $(\theta_2, \theta_3)$ for the forward Euler, probabilistic forward Euler, backward Euler and probabilistic backward Euler methods. Each represents 1000 parameter samples from simulations run with step-sizes $h = 0.005, 0.01, 0.02, 0.05$. This is made of 11000 total samples, with the first 1000 discarded as burn-in, and the remainder thinned by a factor of 10. For each method $M$, its pre-calculated calibration parameter $\alpha_M^*$ is used to set the variance of $\xi$.

At larger step-sizes, the deterministic methods both give over-confident and biased estimates (on different sides of the true value). In accordance with the findings of Conrad et al. (2016), the probabilistic forward Euler method returns a wider posterior which covers the true solution. The

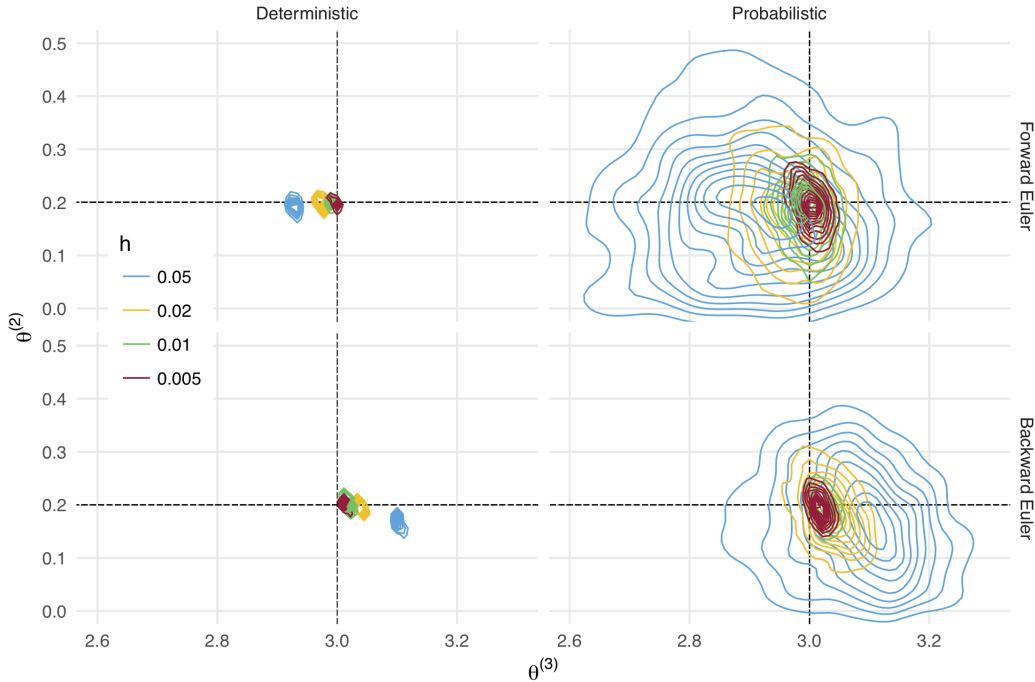

Figure 2: Comparison of the posterior distribution of $(\theta_2, \theta_3)$ from the FitzHugh–Nagumo model in cases where the forward solve is calculated using one of four different integrators (deterministic and probabilistic backward- and forward-Euler methods), each for four different step sizes $h = 0.005, 0.01, 0.02, 0.05$. All density estimates calculated using 1000 MCMC samples. Dashed black lines indicate true parameter values. Full details are given in main text.

bottom right-hand panel demonstrates the same effect with the probabilistic backward Euler method we have introduced in this paper.

We find similar results for second- and higher-order methods, both explicit and implicit. The scale of the effect is however relatively small on such a simple test problem, where a higher-order integrator would not be expected to produce much error in the forward solve. Further work will investigate the application of these methods to more challenging problems.

## 4 Conclusions and avenues for further work

In this paper, we have surveyed the existing collection of probabilistic integrators for ODEs, and proposed a new construction—the first to be based on implicit methods—giving a rigorous description of its theoretical properties. We have given preliminary experimental results showing the effect on parameter inference of the use of different first-order methods, both existing and new, in the evaluation of the forward model. Higher-order multistep methods are allowed by our construction.

Our discussion on integrator calibration does not claim a resolution to this subtle and thorny problem, but suggests several avenues for future research. We have mooted a question on the relationship between the scaling parameter $\alpha$ and other method characteristics. Insight into this issue may be the key to making these types of randomised methods more practical, since common tricks for calibration may emerge which are then applicable to different problems. An interesting direction of enquiry, being explored separately, concerns whether estimates of global error from other sources, eg. adjoint error modelling, condition number estimation, could be gainfully applied to calibrate these methods.

## Acknowledgements

HCL and TJS are partially supported by the Freie Universität Berlin within the Excellence Initiative of the German Research Foundation (DFG). This work was partially supported by the DFG through grant *CRC 1114 Scaling Cascades in Complex Systems*, and by the National Science Foundation (NSF) under grant *DMS-1127914* to the Statistical and Applied Mathematical Sciences Institute's *QMC Working Group II Probabilistic Numerics*.

## Footnotes

[2] We take this opportunity to remind the reader of the unfortunate convention from numerical analysis that results in $s$ having different meanings here and in the previous paragraph—the explicit method of order $s$ is the one with $s$ steps, whereas the implicit method of order $s$ is the one with $s - 1$ steps.

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
