[Supplementary Material · NeurIPS_APPENDIX.pdf]

## Appendix – Proof of Theorem

We first prove statement *(i)*. Let $\omega \in \Omega$ and $x := \xi_i(\omega)$; we will omit the $\omega$-dependence hereafter. Let $v := \Psi_{\theta,s}^h(Z_{i-s+1:i})$. It follows from (8) that

$$z = v + x = Z_i + h\left(\beta_{-1} f(v, \theta) + \sum_{j=0}^{s-1} \beta_j f(Z_{i-j}, \theta)\right) + x. \qquad (12)$$

It follows by rearranging the terms in (12) that

$$\frac{1}{\beta_{-1}}\left(\frac{z - Z_i}{h} - \sum_{j=0}^{s-1} \beta_j f(Z_{i-j}, \theta)\right) = f(v, \theta) + \frac{x}{\beta_{-1} h}.$$

By subtracting $f(z, \theta)$ from both sides, taking the norm, and squaring, we obtain

$$\left\|\frac{1}{\beta_{-1}}\left(\frac{z - Z_i}{h} - \sum_{j=0}^{s-1} \beta_j f(Z_{i-j}, \theta)\right) - f(z, \theta)\right\|^2 = \left\|f(v, \theta) - f(z, \theta) + \frac{x}{\beta_{-1} h}\right\|^2.$$

We may thus rewrite (6) as

$$p(v + x | v, \theta, \eta, h) = \frac{1}{K} \exp\left(-\frac{1}{2\eta^2}\left\|f(v, \theta) - f(v + x, \theta) + \frac{x}{\beta_{-1} h}\right\|^2\right). \qquad (13)$$

and it follows that the normalising constant is given by

$$K(v, \theta, \eta, h) = \int_{\mathbb{R}^d} \exp\left(-\frac{1}{2\eta^2}\left\|f(v, \theta) - f(v + x, \theta) + \frac{x}{\beta_{-1} h}\right\|^2\right) \mathrm{d}x.$$

We now bound the function described in (13) from above and below by unnormalised Gaussian probability densities. First note that by the triangle inequality and the assumption of global Lipschitz continuity we have the lower bound

$$\left\|f(v, \theta) - f(v + x, \theta) + \frac{x}{\beta_{-1} h}\right\| \geq \left\|\frac{x}{\beta_{-1} h}\right\| - \|f(v, \theta) - f(v + x, \theta)\|$$

$$\geq \left\|\frac{x}{\beta_{-1} h}\right\| - L_{f,\theta} \|x\| = \|x\| \left((\beta_{-1} h)^{-1} - L_{f,\theta}\right).$$

Similar reasoning yields the upper bound

$$\left\|f(v, \theta) - f(v + x, \theta) + \frac{x}{\beta_{-1} h}\right\| \leq \|x\| \left(L_{f,\theta} + (\beta_{-1} h)^{-1}\right).$$

Thus for $h > 0$, there exist constants $c_h, C_h > 0$ that do not depend on $\omega$ such that

$$c_h \|x\|^2 \leq \left\|f(v, \theta) - f(v + x, \theta) + \frac{x}{\beta_{-1} h}\right\|^2 \leq C_h \|x\|^2$$

where we have defined $c_h := \left((\beta_{-1} h)^{-1} - L_{f,\theta}\right)^2$ and $C_h := \left((\beta_{-1} h)^{-1} + L_{f,\theta}\right)^2$. Recall that by the definition of the AM method $\beta_{-1} > 0$. (13) now gives

$$\exp\left(-(2\eta^2)^{-1} C_h \|x\|^2\right) \leq K(v, \theta, \eta, h) \cdot p(v + x | v, \theta, \eta, h) \leq \exp\left(-(2\eta^2)^{-1} c_h \|x\|^2\right) \qquad (14)$$

where we note that the lower and upper bounds in (14) do not depend on $v$. In what follows, we will omit the dependence of $p$ and $K$ on $v$ and $\theta$, and write $p_h(\cdot) := p(\cdot | v, \theta, \eta, h)$ and $K_h := K(v, \theta, \eta, h)$, in order to emphasise the dependence of these quantities on $h$.

The interpretation of (14) is that, up to normalisation, the random variable $\xi_i$ has a Lebesgue density that lies between the densities of two centred Gaussian random variables.

Integrating each of the three terms in (14) with respect to $x$ and using the formula for the normalising constant of a Gaussian measure on $\mathbb{R}^d$, we obtain from the hypotheses $\eta = kh^\rho$ and $1 - L_{f,\theta}\beta_{-1}h > 0$ that

$$\left(\frac{\sqrt{2\pi}kh^{\rho+1}\beta_{-1}}{1 + L_{f,\theta}\beta_{-1}h}\right)^d = \left(\frac{2\pi\eta^2}{C_h}\right)^{d/2} =: K_{C,h} \le K_h \le K_{c,h} := \left(\frac{2\pi\eta^2}{c_h}\right)^{d/2} = \left(\frac{\sqrt{2\pi}kh^{\rho+1}\beta_{-1}}{1 - L_{f,\theta}\beta_{-1}h}\right)^d \tag{15}$$

Note that $K_{C,h}$ and $K_{c,h}$ are the normalising constants for the Gaussian random variables $\zeta_{C,h} \sim N(0, (\eta^2/C_h)I_d)$ and $\zeta_{c,h} \sim N(0, (\eta^2/c_h)I_d)$ respectively, where $I_d$ denotes the $d \times d$ identity matrix.

Since $\rho + 1 \ge 0$, the upper and lower bounds in (15) are respectively finite and strictly positive. This proves *(i)*.

To prove *(ii)*, observe that (15) yields that, for all $0 < h < (L_{f,\theta}\beta_{-1})^{-1}$ and $v \in \mathbb{R}^d$, we have

$$1 \le \frac{K_{c,h}}{K_{C,h}} = \left(\frac{C_h}{c_h}\right)^{d/2} = \left(\frac{1 + L_{f,\theta}\beta_{-1}h}{1 - L_{f,\theta}\beta_{-1}h}\right)^d \tag{16}$$

The upper bound decreases to 1 as $h$ decreases to zero, since $L_{f,\theta}$, $\beta_{-1}$ and $h$ are all strictly positive. By the second inequality in (14),

$$\begin{aligned}
\mathbb{E}[\|v + \xi_i\|^2] &= \mathbb{E}[\|Z_{i+1}\|^2] \\
&= \int_{\mathbb{R}^d} \|z\|^2 p(z|v, \theta, \eta, h)\, \mathrm{d}z \\
&\le K_{c,h}K_h^{-1} \int_{\mathbb{R}^d} \|z\|^2 \exp\left(-\frac{c_h\|x\|^2}{2\eta^2}\right)(\mathrm{d}x) \\
&= K_{c,h}K_h^{-1}\mathbb{E}[\|v + \zeta_{c,h}\|^2].
\end{aligned} \tag{17}$$

Since the preceding inequalities hold for arbitrary $v \in \mathbb{R}^d$, we may set $v = 0$ in (13). Using this fact and the fact that (16) implies that $\lim_{h\to 0} K_{c,h}K_h^{-1} = 1$, we only need to show $\mathbb{E}[\|\zeta_{c,h}\|^r] \le C_r h^{(\rho+1)r}$ for some $C_r > 0$ that does not depend on $h$. Consider the change of variables $x \mapsto x' := x(\eta^2/c_h)^{-1/2}$. Since this is just a scaling, we have by the change of variables formula that $\mathrm{d}x = (\eta^2/c_h)^{d/2}\, \mathrm{d}x'$, and hence

$$\begin{aligned}
K_{c,h}^{-1} \int_{\mathbb{R}^d} \|x\|^r \exp\left(-\frac{\|x\|^2}{2\eta^2/c_h}\right)\mathrm{d}x \\
= \left(\frac{2\pi\eta^2}{c_h}\right)^{-d/2} \int_{\mathbb{R}^d}\left[\left(\frac{\eta^2}{c_h}\right)^{r/2}\|x'\|^r \exp\left(-\frac{\|x'\|^2}{2}\right)\left(\frac{\eta^2}{c_h}\right)^{d/2}\right]\mathrm{d}x' \\
= C_r\left(\frac{\eta^2}{c_h}\right)^{r/2} \\
\le C_r' h^{(\rho+1)r}.
\end{aligned} \tag{18}$$

where we have used (15) in the first equation, and where $C_r, C_r' > 0$ do not depend on $h$.

To prove *(iii)*, we set $r = 2$ and $\rho \ge s + \frac{1}{2}$ in *(ii)* to obtain $\mathbb{E}[\|\xi_i\|^2] \le ch^{(2s+3)}$. Since $s$ is the number of steps of the Adams–Moulton method of order $s + 1$, the random variable $\xi_i^h$ satisfies the assumption in the statement of Theorem 3 in Teymur et al. (2016). It then follows from that result that

$$\sup_{0 \le ih \le T} \mathbb{E}\|Z_i - x(t_i)\| \le ch^{2(s+1)}$$

(Note: the $s$ used here is different to $s$ in the referenced paper, since we follow the usual convention in numerical analysis texts where the implicit multistep method of order $s$ has $s - 1$ steps.) $\qquad\square$

# Appendix – MCMC Psuedo-code

| Algorithm for sampling $p(\theta, Z\|Y)$ |
|---|

1   INPUT $\theta^{[1]}$
2   $\xi^{[1]} \sim p(\xi)$
3   FOR $1 \leq k \leq K$
4       $\phi^{[k,k]} \leftarrow p(Y\|Z,\sigma)p(Z\|\theta^{[k]},\xi^{[k]})p(\theta^{[k]})$
5       $\theta^* \sim q(\cdot\|\theta^{[k]})$
6       $\phi^{[*,k]} \leftarrow p(Y\|Z,\sigma)p(Z\|\theta^*,\xi^{[k]})p(\theta^*)$
7       $\alpha^{[k]} \leftarrow \min(1, \phi^{[*,k]}/\phi^{[k,k]})$
8       $r^{[k]} \sim \mathcal{U}[0,1]$
9       IF $r^{[k]} < a^{[k]}$
10          $\theta^{[k+1]} \leftarrow \theta^*$
11          $\xi^{[k+1]} \sim \mathbb{P}_\xi$
12          $k \leftarrow k+1$
13      ELSE
14          $\theta^{[k+1]} \leftarrow \theta^{[k]}$
15          $k \leftarrow k+1$
16      END
17  END
18  OUTPUT $\theta^{[2]}, \ldots, \theta^{[K]}$