[Reviews · NeurIPS 2018]

Reviewer 1



Overview: This paper builds on and adapts the explicit randomized integrators developed by Conrad et al. (2016) and Teymur et al. (2016) and proposes a alternative sequential procedure that directly incorporates information from the ODE at the next time step to produce an implicit probabilistic integrator (equation (6)). They give rigorous description of its theoretical properties and extend the method to multi-dimensional case. They also provide some preliminary experimental results on a two-dimensional non-linear dynamic system with three parameters. The results show their proposed method showed a clear de-biasing effect, which is not discovered by Conrad et al. (2016). Quality: I’m not quite sure if the paper is technically sound (I don’t see how the equation (5) together with higher order implicit linear multistep methods of Adams-Moulton leads to equation (6)). I did not thoroughly check all the equations (especially the ones in the supplementary materials) though. Clarity: This paper is not very easy to follow probably due to the heavy notation in math symbols. I also found some statements are confusing. For example, section 1 (line 28) the authors state that they focus on the initial value problems (IVPs) Originality: The proposed method seems to be novel. Significance: Although this paper provides an interesting direction in solving the ODE problems, the content of experiment section seems to be premature for NIPS publication. I suggest to condense the related work session and incorporate more results in the experiment session, and possibly explore the relationship between the scaling parameters and other method characteristics. ------ Post rebuttal: Thanks for the clarification for equation (5) and (6). It totally makes sense now. I do have a minor suggestion: In line 158, it says "r is a positive \eta-scaled discrepancy measure ...", however on line 169, it says "r(z) is the expression ... and g is in this case the transformation g: ...", the expression of r(z) does not contain \eta and "g" on the other hand contains \eta. It'll make it easier to follow if the definition of r and g is consistent. How about changing g(r(z, \eta)) to (g(r(z), \eta), and explicitly stating that r(z) = F_{i+1)(z) - f(z, \theta)? I appreciate the details and clear description in the literature and agree that it provides a good context for the new work. Although I still think it should be shortened to leave space for more experiments, the novelty of the paper is still worth to be published.

Reviewer 2



## Summary The paper proposes a novel algorithm for solving initial value problems in ODEs. The first part of the paper aims to review current literature on probabilistic ODE solvers arising in the field of probabilistic numerics (PN). Then, based on this review, the authors propose a novel probabilistic implicit solver for IVPs which retains some of the non-linearities of the vector field, but locally works with Gaussian approximations to the solution and/or the velocities for computational speedups. The paper contains a proof of convergence in mean square for diminishing step sizes. As a proof-of-concept, the authors demonstrate their solver as well as its application to system identification on a 2D-toy-problem. ## Novelty & originality & Significance The paper is an interesting read since it is a combination of a review paper and a new proposed method. Apart from the latter, personally I think it has a high values since it summarizes the state-of-the-art of the field well, also for non-ODE experts. Additionally the proposed method (*implicit* rather than explicit PN solver) seems novel to me. It is an interesting approach to handling the cost-precision tradeoff for ODEs which more generally is inherent to any numerical solver. I am not super convinced about the practical relevance of the method in its current state and I’d encourage the authors to comment on the possible applications, especially in contrast to running a faster solver with a smaller step h for the same computational budget. ## Clarity The paper is clearly written and well developed. It is however very dense. I’d actually like to see a longer journal version of this, too. ## Major points - Eq. 3 and paragraph below. I do not see the point why Z, Z-tilde and x are named differently since, as I understand, all three represent the same random variable (the unknown solution). - Eq. 4: the authors mention that their method could in principle use different transformations g but only the Gaussian is used. Did the authors experiment with other transformations? If yes, which? If not, do the authors think that for all practical purposes one might always be restricted to use Gaussian forms of g? - a general point on Gaussians: the authors seem to point out repeatedly that some distributions are non-Gaussian. This is fair enough to distinguish their method from methods that do use Gaussians in all distributions. But for a reader it is sometimes harder to understand the novel method by having negative statements only (what it is not, rather than what it is). I’d encourage the authors to also point out more clearly which of the latent variables (F for instance? Or r?) are Gaussian locally, especially since Gaussians seem to appear later during the approximations and estimation of H. It would make for an easier read. - l. 242 “… nowhere assuming that the distributions are Gaussian.” Related to the above point, this comment is a bit misleading and prone to misunderstandings since it is not clear what “the distributions” means. I’d encourage the authors to specify which distribution they were referring to, especially since Gaussian distributions do occur in various places. - l. 270. “Then non-linearities from f are retained (to second order)” I am unsure, but isn’t the approximation linear in f but non-linear in z? At least l. 265 seems to indicate this. - l. 271 “jointly Gaussian”. What is meant by “jointly” here? Is it jointly across iterations i? Or jointly given one specific i? The former is true, the latter is not I guess. Could the authors please specify. - Eq. 10: H seems to be fitted in a kind of empirical Bayes. Could the authors comment on this, please. Especially in connection to the idea of “numerical data F”. - A general point on applicability: As a potential user of the algorithms, what would be the application where the proposed algorithms is better than any existing ODE solver. And also better in which sense? I am asking because the authors motivate their algorithms also by the accuracy-cost tradeoff, and it left me wondering what I would actually use it for. I could, for example also just run a classic ODE solver with a finer step size h and, in the same time, have such a good approximation to my solution that I do not care very much about potential non-symmetric error bars. - Experiments: please explain which of the approximations are used to generate the results. For instance I think the authors did not mention if the approximations of section 2.4 are used. Also how does the sampling exactly work. Could the authors please add a pseudo-code to the supplements. - Figure 1 & experiments: Once given the empirical distribution (red dots), is it possible to generate an estimator for the solution? If yes, how? What would be the most favorable choice and why? ## Minor points - l. 101 and l. 135 “This approach…” Both times a new paragraph is started with a reference to something mentioned before. Possibly the paragraphs can be divided somewhere else such that paragraphs contain one thought process completely. Then tracing back and referencing would become a lot easier. - Figure 2. Could the authors please add an indication of the true value to the x-axis? Like this one has to trace-back to find the true value of theta in the text. ## Post rebuttal Thank you for clarifying and addressing my concerns, especially also all the detailed comments [3]. [1] purpose of work: thank you for clarifying. That makes sense to me, although my question was more targeted towards specific examples or applications where, as you say, the richer UQ is preferred over tighter worst-case errors. I understand the high-level argument but would be interested in knowing an application where this trade-off is indeed in favor of the PN-ODE solver. If the authors are aware of one it might be worth mentioning it in the final version. In my mind, a less well-understood but small error is usually preferred over a well-understood, but larger error. [2] notation Schober et al.: I see your point here. I’d still argue that it is worth mentioning that Z and \tilde{Z} represent the same RV and the distinction in notation only emphasizes the difference in their use. I am kind of picking on this since distinguishing the two is a bit counter-intuitive from a PN point of view and possibly misleading. [3] all other questions: thanks for clarifying and addressing all the points. It is clearer now.

Reviewer 3



The authors discuss the recent literature of probabilistic numerical (PN) methods for solving initial value problems (IVPs). Based on this analysis, the authors propose a novel probabilistic model for IVPs whose structure resembles implict methods for IVPs. The authors present a rigorous error analysis and compare their method on a standard problem in the literature. PN is an idea that seems to generate momentum in the community (two previous NIPS papers, an AISTATS paper, an UAI paper and further journal works for ODE solvers alone). In terms of clarity, a strength of this work is that it presents an excellent overview and comparison of related work. I particularly appreciated the context to gradient matching methods. This clarity continues in the derivation of their own work, as well as in their detailed discussion of the experiment. This clarity and detail of description is the flip side of the weakness of the paper: it only provides one experimental evaluation. In particular, I would have liked to see a discussion of the solver for linear problems for which the implicit likelihood (6) turns into a non-central Gaussian. This would have been particularly interesting, as classical implicit methods are highly effective for stiff problems. It could have been investigated whether this holds for their probabilistic counterparts as well. Secondly, I had trouble following the description of Sect. 2.4. In particular: did I understand correctly, that you do _not_ sample from (6) in practice, but from the non-centered Gaussian described in Sect. 2.4? How does this not introduce a bias into the method? As a side note: if I determine a valid Lipschitz constant for f, I should be able to generate a rejection sampler zeta_c,h, correct? I think this work could be an important piece in the puzzle of covering all variations of classical numerical integrators. As the authors rightly point out: many previous publications on the subject have thoroughly, but narrowly, explored the space of explicit step-by-step methods. I have the impression that the path followed by the authors here could lead to a whole new class of probabilistic numerical integrators that might have properties orthogonal to the existing class of solvers. Thus, I strongly recommend this work for acceptance. As some open questions, in particular the analysis for linear and stiff problems, remain completely unanswered, I refrain from giving a higher score. Minor points for improvement: - In Sect. 1.1, the authors could consider also discussing https://arxiv.org/abs/1803.04303 - The last sentence of Sect. 1.2 mentions the connection to predictor-corrector methods. Schober et al. (2018) point out that their model corresponds to an implicit method, although they only evaluate it in PEC fashion. This connection could have been discussed in your work. - In Sect. 2.3, the authors could consider refering to https://arxiv.org/abs/1702.03673 - In the discussion of the scaling matrix H below Eq. (10), the authors could discuss the resemblance of their idea to the Milne device of predictor-corrector methods - I have a mixed impression of the comments on the "de-biasing effect" (starting on line 331). I appreciate that they highlight the observation and I agree with their rationale on the method by Conrad et al., but I think it's a bit too early to attribute a de-biasing effect to the novel method based on one experiment. In this respect, the authors might also want to discuss the connection to probabilistic symplectic integrators of https://arxiv.org/abs/1801.01340 Some stylistic feedback which is orthogonal to the scientific discussion. a) the authors often apply dashes in their punctuation. I suggest using an em-dash and dropping the neighboring white-space like---so. https://en.wikipedia.org/wiki/Dash#Em_dash b) Figure 2: the authors could mark the true value of theta_3 = 3.0 in the figure and also put it in the caption. --- Post-rebuttal update: I have read the rebuttal and agree with the authors' points. My overall (already good) impression from the paper was unfortunately not further improved, but I appreciate the clarifications.